# Serial Llama Immunization with Various SARS-CoV-2 RBD Variants Induces Broad Spectrum Virus-Neutralizing Nanobodies

**DOI:** 10.3390/vaccines12020129

**Published:** 2024-01-26

**Authors:** Pavel P. Solodkov, Alexander M. Najakshin, Nikolai A. Chikaev, Sergey V. Kulemzin, Ludmila V. Mechetina, Konstantin O. Baranov, Sergey V. Guselnikov, Andrey A. Gorchakov, Tatyana N. Belovezhets, Anton N. Chikaev, Olga Y. Volkova, Alexander G. Markhaev, Yulia V. Kononova, Alexander Y. Alekseev, Marina A. Gulyaeva, Alexander M. Shestopalov, Alexander V. Taranin

**Affiliations:** 1Institute of Molecular and Cellular Biology Siberian Branch of the Russian Academy of Sciences, 630090 Novosibirsk, Russia; solodkov.pavel@mcb.nsc.ru (P.P.S.); najakshin@mcb.nsc.ru (A.M.N.); na_chik@mcb.nsc.ru (N.A.C.); lucie@mcb.nsc.ru (L.V.M.); baranov@mcb.nsc.ru (K.O.B.); sergey.v.guselnikov@gmail.com (S.V.G.); ochotanya@gmail.com (T.N.B.); chikaev@mcb.nsc.ru (A.N.C.); volkova@mcb.nsc.ru (O.Y.V.); 2Federal Research Center of Fundamental and Translational Medicine, 630117 Novosibirsk, Russia; mag87@ngs.ru (A.G.M.); yuliakononova07@yandex.ru (Y.V.K.); al-alexok@yandex.ru (A.Y.A.); mgulyaeva@gmail.com (M.A.G.); shestopalov2@mail.ru (A.M.S.); 3Department of Natural Sciences, Novosibirsk State University, 630090 Novosibirsk, Russia

**Keywords:** SARS-CoV-2, COVID-19, coronavirus variants, viral evasion, broadly neutralizing antibody, single-domain antibody, antiviral nanobody

## Abstract

The emergence of SARS-CoV-2 mutant variants has posed a significant challenge to both the prevention and treatment of COVID-19 with anti-coronaviral neutralizing antibodies. The latest viral variants demonstrate pronounced resistance to the vast majority of human monoclonal antibodies raised against the ancestral Wuhan variant. Less is known about the susceptibility of the evolved virus to camelid nanobodies developed at the start of the pandemic. In this study, we compared nanobody repertoires raised in the same llama after immunization with Wuhan’s RBD variant and after subsequent serial immunization with a variety of RBD variants, including that of SARS-CoV-1. We show that initial immunization induced highly potent nanobodies, which efficiently protected Syrian hamsters from infection with the ancestral Wuhan virus. These nanobodies, however, mostly lacked the activity against SARS-CoV-2 omicron-pseudotyped viruses. In contrast, serial immunization with different RBD variants resulted in the generation of nanobodies demonstrating a higher degree of somatic mutagenesis and a broad range of neutralization. Four nanobodies recognizing distinct epitopes were shown to potently neutralize a spectrum of omicron variants, including those of the XBB sublineage. Our data show that nanobodies broadly neutralizing SARS-CoV-2 variants may be readily induced by a serial variant RBD immunization.

## 1. Introduction

Since the onset of the coronavirus pandemic in 2019, significant research efforts have been focused on developing preventative and therapeutic measures based on monoclonal SARS-CoV-2-neutralizing antibodies. Thousands of human antibodies targeting SARS-CoV-2 Spike have been identified and characterized by research groups and companies worldwide [1,2]. Some have been authorized for emergency use in high-risk patient populations. However, at present, no effective anti-COVID therapeutic antibodies are available due to the emergence and spread of novel mutant variants of SARS-CoV-2 capable of evading the immune response induced by the original Wuhan strain of the virus. Even early mutant variants, such as Beta and Delta, exhibited resistance to some of the approved therapeutic antibodies, and the emergence of the Omicron lineage in November 2021 rendered most of them essentially inactive [3,4]. Unlike earlier mutant variants of the virus, which differed from the Wuhan strain by single substitutions, the first Omicron lineage variant, v.1.1.529, had 30 Spike protein mutations, including 15 in the receptor-binding domain (RBD). By 2023, the XBB sub-lineage of Omicron had gained widespread prevalence and was responsible for over 90% of documented SARS-CoV-2 infections worldwide at the time of writing. Various sublineage variants, such as XBB, XBB.1, XBB.1.5, and XBB.1.16, among others, demonstrated high resistance not only to serum antibodies from vaccinated or previously infected individuals but also to all the authorized therapeutic antibodies [5,6,7].

In addition to classical human and animal antibodies, significant attention has been given to the development of antiviral nanobodies during the pandemic. This antibody format, composed of VHH domains from camelid heavy-chain-only immunoglobulins, has several important advantages, including high stability, low immunogenicity in humans, ease of constructing oligomers, and the ability to recognize epitopes inaccessible to classical antibodies [8,9]. Given current insights into mechanisms of virus immune evasion [10,11], it is conceivable that at least some of the nanobody epitopes are less prone to mutation escape compared to those of the classical antibodies. Numerous studies published over the past three years have shown that nanobodies and nanobody-based multivalent proteins may display remarkable SARS-CoV-2-neutralizing properties [12,13,14,15,16,17,18,19]. However, the question of whether nanobodies have any advantages in terms of the breadth of neutralization compared to the classical antibodies remains open. Xiang et al. have demonstrated that successive immunization of a camelid by Wuhan variant RBD can enhance the development of pan-sarbecovirus nanobodies [16]. Ketaren et al. [20] and Ma et al. [18] reported that some of the nanobodies derived from the immune repertoire elicited against the Wuhan variant of the virus were effective against some of the new variants, including early Omicron representatives. However, none of these nanobodies alone exhibited the ability to neutralize the entire spectrum of viral variants. To our knowledge, neither of the Wuhan SARS-CoV-2 induced nanobodies were shown to potently neutralize XBB sublineage variants.

Earlier, we described a panel of highly potent monoclonal human antibodies that neutralized the Wuhan variant of SARS-CoV-2 and several early coronavirus variants [21,22]. As a parallel research direction, we also investigated the properties of SARS-CoV-2-specific llama nanobodies. During this study, two phage libraries of RBD-specific nanobodies were constructed. In one library obtained in 2020, the VHH repertoire of a llama after immunization with the ancestral Wuhan variant of RBD was used. In the other library, constructed at the end of 2022, the VHH repertoire of the same llama was probed following additional sequential immunizations with Beta, Delta, SARS-CoV-1, and Omicron (BA.5) RBD variants. The nanobodies discovered in the first library effectively neutralized the original SARS-CoV-2 variant both in vitro and in vivo but lacked activity against Omicron lineage variants. Screening of the second library led to the identification of clones with a significantly broader spectrum of activity, including the ability to neutralize the lentiviral particle variants such as BQ.1, XBB, XBB.1, XBB.1.5, and XBB.1.16, which were not used during immunization.

## 2. Materials and Methods

### 2.1. Llama Immunization

Llama (*Llama glama*, adult female) was immunized in several steps according to the standard protocol using Fama adjuvant (Gerbu, Heidelberg, Germany) [23]. First, in early 2020, 4 weekly subcutaneous injections of 400 µg of RBD Wu reconstituted in 150 mM NaCl were given. Throughout the course of immunization, the immune response was monitored using ELISA. Once the titer of RBD-specific antibodies rose above 1:100,000, peripheral blood lymphocytes were isolated from a 100 mL blood sample, drawn 7 days after the last immunization, via a Ficoll-Paque gradient yielding about 10^8^ B cells. RNA was isolated from these cells, converted into cDNA, and used to construct library 1. Next, this llama was immunized with Beta and Delta RBD variants in the second half of 2021 (2 injections), SARS-CoV-1 RBD in early 2022 (4 injections), and BA.5 RBD in the second half of 2022 (4 injections). At the end of the immunization, peripheral blood lymphocytes were isolated, and the material was used to construct library 2.

### 2.2. Expression and Purification of SARS-CoV-2 RBD Antigen

His-tagged RBD of the Wuhan variant was described previously [21]. PCR fragments encoding Strep-tagged RBD of various mutant variants were amplified from the corresponding Spike protein encoding plasmids using the primers rbdVAgeF (5′-GCACCGGTAGAGTGCAGCCCACCGAATCCATC-3′) or rbdVXmaF (5′-GCCCCGGGAGAGTGCAGCCCACCGAATCCATC-3′) and rbdVBamR (5′-ACGGATCCCGAAGTTCACGCATTTGTTCTT-3′). PCR fragments and the pCDH3 vector encoding a C-terminal Strep-tag were digested with AgeI or XmaI and BamHI (SibEnzyme, Novosibirsk, Russia) and ligated together. All the RBD variants, therefore, have a Strep-tag at their C-terminus and were expressed in HEK293T cells as described earlier for His-tagged RBD [21] and affinity purified on a Strep-Tactin^®^ TACS agarose (IBA Lifesciences, Göttingen, Germany). NaCl was added to cell supernatants to 150 mM, and pH was adjusted to 8.0 using 1.5 M Tris-HCl pH 8.9. Next, the solution was loaded onto a 3 cm^3^ streptactin-agarose column, which was then washed with PBS until minimal values of OD_280_ were achieved. To elute the bound RBD proteins, 2.5 mM D-desthiobiotin (Sigma-Aldrich, St. Louis, MO, USA, D1411) in PBS was added to the column. This was followed by sample dialysis and biotinylation, as described for His-tagged RBD.

### 2.3. Construction of Nanobody Libraries for Phage Display

Total RNA was isolated essentially as described by Chomczynnski and Sacchi [24] and used as a template for cDNA synthesis. First-strand synthesis of cDNA from the resulting total RNA was prepared using the RevertAid First Strand cDNA Synthesis Kit (Thermo Fisher Scientific, Waltham, MA, USA) with oligo(dT)_18_ and 5 µg of total RNA used as an input. Nanobody sequences were amplified with nested PCR and cloned into a phagemid vector pHEN4HH for expression as pIII fusion [25]. In brief, the first PCR was performed with the primers CALL001 (5′-GTCCTGGCTGCTCTTCTACAAGG-3′) and CALL002 (5′-GGTACGTGCTGTTGAACTGTTCC-3′). Two main PCR products were obtained and separated on an agarose gel. The shorter PCR fragment (700 bp) was cut out and extracted using the QIAquick Gel Extraction Kit (Qiagen, Hilden, Germany). After reamplification of the VHH gene fragments with nested primers (VHHback- 5′-GATGTGCAGCTGCAGGAGTCTGGRGGAGG-3′ and PMCF 5′-CTAGTGCGGCCGCGTGAGGAGACGGTGACCTGGGT-3′) annealing at the framework 1 and framework 4 regions, the resulting PCR products were digested with PstI and NotI and ligated into the pHEN4HH phagemid vector (cut with the same restriction enzymes). Phagemid pHEN4HH was created by cloning the NotI-EagI fragment encoding the His_6_ sequence into pHEN4 [26]. Recombinant phagemids were transferred into *E. coli* TG1 cells (Agilent, Technologies, Santa Clara, CA, USA) via electroporation. A VHH library of about 1 × 10^8^ independent transformants was obtained. The percentage of recombinant nanobody-encoding phagemids was ~95%, as assessed by colony PCR using the MP57 (5′-TTATGCTTCCGGCTCGTATG-3′) and GIII (5′-CCACAGACAGCCCTCATAG-3′) primers on 48 randomly chosen colonies. The resulting TG1 library stock was then infected with VCSM13 helper phage to obtain a library of VHH-presenting phages. The rescue with helper phage VCSM13 and polyethylene glycol precipitation was performed as described previously [27]. Specifically, *E. coli* TG1 cells grown in 2xYT medium supplemented with 50 µg/mL ampicillin and 2% glucose at 37 °C were inoculated in log-phase with VCSM13 helper phages and incubated at 37 °C for 30 min without shaking. Afterward, the bacterial culture was centrifuged at 2800× *g* for 10 min, the medium was replaced by 2xYT containing 50 µg/mL ampicillin and 25 µg/mL kanamycin, and the culture was incubated overnight under shaking at 37 °C. Subsequently, the material was centrifuged, and the supernatant was PEG precipitated (20% PEG 6000 in 2.5 M NaCl) at 4 °C for 1 h. Phages were spun down, and the pellet was resuspended in 1 mL PBS.

### 2.4. Phage Library Biopanning

The nanobodies binding to SARS-CoV-2 RBD were selected by phage display. Briefly, 60 µg biotinylated RBD was immobilized onto 20 µL MagnaBind Streptavidin (Thermo Fisher Scientific, Waltham, MA, USA) in PBS. After washing the beads with PBST (PBS containing 0.1% Tween20), the phage library (10^11^ phages/selection round) and the beads (bioRBD-coated or naked) were pre-blocked in PBS/milk 2% (*w*/*v*) and allowed to associate for 60 min at room temperature with mixing. Unbound phages were removed by 6 washes with PBST and 2 washes with PBS. The elution of phage particles bound to the RBD-coated magnetic beads was performed with a 10 mM DTT solution. Aliquots of the eluted phage suspensions from bioRBD-coated or naked beads were used to infect *E. coli* TG1 cells for evaluating the enrichment, which was at least 100-fold (bioRBD vs. naked beads). Eluted phages were then used to infect exponentially growing *E. coli* TG1 cells in order to amplify and prepare phage particles for the next selection round. For library 1, one panning round with RBD Wu as the target was performed, whereas library 2 was panned once against BQ.1.1, followed by the XBB.1.5 RBD.

### 2.5. Immunoaffinity Competition-Based Enrichment of the Phage Library

To enrich the phage library with neutralizing clones, we followed the protocol proposed by Esparza T.J. et al. with minor modifications [28]. Recombinant His6-SUMOstar-ACE2 [21] diluted to 1 µg/mL in 0.1 M NaHCO_3_, pH 9.6, was immobilized in 5 wells of a flat-bottom strip at 4 °C overnight. The solution was then removed and blocked with 2% non-fat dry milk in 0.1 M NaHCO_3_, pH 9.6. Amplified phage clones (~10^8^ phages) were next pre-incubated with 60 ng of biotinylated RBD (Wuhan variant for library 1, XBB.1.5 for library 2) for 30 min at room temperature. Phage/bio-RBD mixture was added to the first well and kept on a shaker for 30 min at room temperature. The mixture was then transferred to the neighboring well and incubated as described above. Phages were collected from the fifth well, mixed with 10 µL of MagnaBind Streptavidin, and incubated on a mixer for 15 min at room temperature. Magnetic beads were washed 5 times with PBS, and phages were eluted in 10 mM DTT; infectivity was then measured to assess the degree of enrichment.

### 2.6. Phage Display Clone Screening

After panning, individual bacterial colonies were randomly picked in a 96-deep well block, grown in 400 μL of 2xYT medium supplemented with 0.1% glucose and 100 μg/mL ampicillin until OD_600_ = 0.6 and VCSM13 helper phage (10^9^ phages/well) was added to the transformed bacteria for release of phage particles encoding VHH. The culture was kept stationary for 30 min at 37 °C, and then kanamycin was added at the final concentration of 50 μg/mL. After 16 h of incubation with shaking at 37 °C, the cells were pelleted by centrifugation at 2800× *g* for 10 min, and the supernatant was used for ELISA. Nunc Maxisorp 96-well plates were coated with 100 μL of RBD solution at 1 μg/mL in 0.1 M NaHCO_3_, pH 9.6 overnight at 4 °C. The coating solution was removed, and the plate was blocked with a 0.1 M NaHCO_3_, pH 9.6, 2% (*w*/*v*) non-fat dry milk solution. Each phage supernatant was added to a well coated with RBD and another well without and incubated for 1 h at room temperature. The assay plate was washed, and the antigen-bound phages were detected with 1/2000 diluted mouse anti-M13 IgG (Sino Biological, Wayne, PA, USA, RRID: AB_2857926) and 1/2000 rabbit anti-mouse IgG conjugated with HRP (Sigma-Aldrich, St. Louis, MO, USA, RRID: AB_258431). Plates were washed five times in wash buffer and HRP, and activity was determined by adding 100 μL/well substrate solution (0.5 mg/mL o-Phenylenediamine dihydrochloride, 0.05 M phosphate-citrate buffer, pH 5.0, 0.01% hydrogen peroxide) with 1–3 min incubation and the reaction was quenched by adding 100 µL of 30% H_2_SO_4_. Plates were read at 491 nm on a Multiskan FC microplate reader (Thermo Fisher Scientific, Waltham, MA, USA). The colonies corresponding to the wells that showed positive signals (if the optical density 491 nm was greater than two standard deviations above the background) in ELISA were grown and used to prepare phagemid DNA and sequencing.

### 2.7. Expression and Purification of Nanobodies

The vector DNA from selected positive clones was transformed into the non-suppressor strain HB2151 of *E. coli* (Agilent, Technologies, Santa Clara, CA, USA). The cells harboring the recombinant phagemids were grown at 37 °C in 400 mL LB-ampicillin, 0.1% glucose, 1 mM MgCl_2_ in culture flasks until OD_600_ = 0.7 was reached. Then, expression was induced with 1 mM IPTG, the temperature was lowered to 28 °C, and cells were grown overnight for an additional 14–15 h. Cells were then harvested by centrifugation at 9000× *g* for 15 min. To prepare periplasmic extracts, the cell pellet was resuspended in 10 mL of ice-cold TES (0.2 M Tris-HCl, pH 8.0, 0.5 mM EDTA, 0.5 M sucrose) and incubated for at least 1 h on ice with shaking. Then, 20 mL TES/4 buffer was added, and the cell suspension was shaken on an orbital shaking platform for 45 min on ice. The cells were pelleted by centrifugation for 30 min at 10,000× *g* at 4 °C, and the supernatant was loaded on a 1 mL Ni-NTA column. The column was washed with 20 mL of buffer A (20 mM sodium phosphate, pH 7.4, 300 mM NaCl, 5 mM Imidazole), and then the protein was eluted with buffer B (20 mM sodium phosphate, 300 mM NaCl, 300 mM Imidazole) and dialyzed against PBS buffer. After elution from Ni-NTA, protein purity was usually above 90%, as judged by SDS-PAGE. To express the protein at a large scale, *E. coli* Shuffle T7 Express cells were transformed with pE-SUMO-VHH-Avitag plasmid obtained by inserting a synthetic fragment encoding an individual nanobody sequence and Avitag^TM^ (GLNDIFEAQKIEWHE) into pE-SUMO-kan (LifeSensors, Malvern, PA, USA). A single colony was used to inoculate 3 mL LB (kanamycin, 50 µg/mL). Following overnight growth at 37 °C, the 5 mL culture was used to inoculate 1000 mL TB (kanamycin, 50 µg/mL), and the cells were grown at 30 °C with shaking. At an OD_600_ = 0.7, IPTG was added to 1 mM, and the cell continued to grow overnight at 16 °C and vigorous shaking. Cells were harvested, resuspended in 50 mL of buffer A with 1 mM PMSF, 0.5 mM EDTA, pH 8.0, and lysed by sonication. Cell debris was pelleted by centrifugation for 30 min at 25,000× *g*, and the supernatant was filtered through a 0.22 μm PES filter (TPP). The solution was loaded on a 3 mL Ni-NTA column, and the protein was purified as described above. After elution from the column, the protein was cleaved by SUMO-protease (isolated using pFGET19_Ulp1 plasmid #64697, Addgene, Watertown, MA, USA). The sample was first dialyzed against reaction buffer (25 mM Tris, pH 8.0, 300 mM NaCl, 1 mM DTT) overnight at 4 °C, then ~1000 units (according to a protein amount) of SUMO-protease was added to the dialysis tubing and incubated for 12 h at 4 °C in fresh buffer. Imidazole was added to 30 mM, and the dialyzed sample was loaded onto a 1 mL Ni-NTA resin equilibrated in Buffer A. The flow-through containing purified nanobody was collected and concentrated using Amicon Ultra-15 Ultracel-3 (Millipore, Burlington, MA, USA) to a final concentration of 1 mg/mL.

### 2.8. Surrogate Assays for Competitive Nanobody Binding to SARS-CoV-2 RBD

Competition ELISA was performed to select the nanobodies capable of blocking the interaction between RBD and ACE2. First, Nunc 96-well Maxisorp plates were used to immobilize His6-SUMOstar-ACE2 fusion at 2 µg/mL in 0.1M NaHCO_3_, pH 9.6, overnight at 4 °C. The next day, the solution was removed, and the wells were washed 5 times with 0.1 M NaHCO_3_, pH 9.6, 0.1% Triton X-100, 2 min/wash. Then, the plate was blocked with a 0.1 M NaHCO_3_, pH 9.6, 2% (*w*/*v*) non-fat dry milk solution. In parallel, 50 µL of nanobodies at appropriate concentrations were mixed with 50 µL biotinylated RBD at 0.2 µg/mL. 100 µL of nanobody/RBD pre-mixes were then added to the wells and allowed to stay for 2 h at room temperature. Wells were then washed 5 times with 0.1 M NaHCO_3_, pH 9.6, 0.1% Triton X-100, 5 min/wash, and 100 µL of Streptavidin-HRP conjugate (1:1000) (Sigma-Aldrich, St. Louis, MO, USA, #RABHRP3) was added. Following incubation for 1 h at room temperature, the wells were washed as above (5 times, 5 min/wash). Next, 100 µL of substrate solution was added and incubated for 1–3 min until color developed, and the reaction was quenched with 100 µL of 30% H_2_SO_4_. Plates were then read at 491 nm on a Multiskan FC microplate reader (Thermo Fisher Scientific, Waltham, MA, USA). FACS-based assay for competitive nanobody binding to SARS-CoV-2 RBD (ACE2 blocking cell-based assay) was essentially described earlier [21].

### 2.9. Measurements of Epitope Binning (Competition Assay) and Binding Kinetics and via BLI

These methods were performed as described previously [21] with minor modifications. ForteBio Octet RED96e system (Sartorius, Göttingen, Germany) was used. For binding kinetics measurements, Nbs (10 μg/L) were immobilized on the sensor chips, and RBD Wu was added at 5 concentrations (2.5 nM, 5 nM, 10 nM, 20 nM, and 40 nM). The durations of the following steps were modified: Association 500 s, Dissociation 1000 s. For epitope binning, RBD-His6 Wu (10 μg/mL) was immobilized on the sensors. Nb1, Nb2 at 50 nM, and Nb1-Fc and Nb2-Fc at 500 nM were added. The following steps were modified: Loading 300 s, Saturation Nb1 (Nb1-Fc) 500 s, Competing (Nb2-Fc) 500 s, Activation with 10 mM NiCl_2_ 60 s. Multistep competition analysis was performed by following scheme: Baseline 60 s, Loading 9A57-SUMO through 6His tag (10 μg/mL) for 300 s on NTA biosensor (Sartorius, Göttingen, Germany), Baseline2 60 s, sequential saturation with RBD Wu, 9A7, 9A21, and 9A58 (200 nM each) for 200 s.

### 2.10. Expression and Purification of Nb-Fc Fusions

To obtain homodimers of nanobodies with Fc fragment, sequences encoding single-domain antibodies H5, G7, 2F4, 9A7, 9A21, 9A57 and 9A58 were amplified using primers Vhhfor (3′-GATTCACCGGTGTACATTCTCAGGTGCAGCTGCAGG-5′) and VhhIgrev (3′-CATGGGCCCGAGGAGACGGTGACCTGG-5′) from the pHEN4HH vector and cloned at AgeI-ApaI restriction sites into the eukaryotic pCDH3 expression vector containing the sequence of the human IgG1 Fc. The expression and purification protocol was the same as for Ab [21], but Amicon Ultra-15 Ultracel-30 was used to concentrate the samples.

### 2.11. SARS-CoV-2 S-Pseudotyped Lentivirus Neutralization Assay

SARS-CoV-2 S-pseudotyped lentiviral particles were produced as described previously by transfection of HEK293T cells with a 4:6:3 molar mixture of plasmids psPAX2, pCDH-NLuc, and a pCAGGS-SpikeD19 plasmid encoding either a Wuhan or a mutant variant of the SARS-CoV-2 S protein [21]. The constructs encoding mutant variants of the Spike protein were made using either gene synthesis (Genewiz, South Plainfield, NJ, USA, or Eurogene, Russia) or mutagenesis with sets of mutagenic primers. Sanger sequencing was used to confirm the sequence identity. Neutralization assays for H5, G7, 2F4, and their Fc fusions were conducted in the same way as described by Gorchakov et al. [21]. For 9A7, 9A21, 9A57, and 9A58, their Fc fusions, 9A4 and 9A80, a modified version of the assay was used [22]. 48 h following transduction of the ACE2-HEK293N cells with a mixture of antibody and S-pseudotyped lentivirus, the cells were washed with PBS and lysed in PBS with 0.2% Triton X-100. ACE2-HEK293T cells transduced in the absence of antibodies or non-transduced cells were used as the controls. Luminescence intensity was measured by Luminoscan (Thermo Fisher Scientific, Waltham, MA, USA) 100 ms after the addition of the substrate to the well (1.25 µg of h-coelenterazine in 50 µL of PBS) over the period of 3 s. Integral fluorescence was used for further calculations. The half-maximal inhibitory concentration (IC_50_) was determined by non-linear regression as the concentration of nanobody or nanobody-Fc fusion that neutralized 50% of the pseudotyped lentivirus.

### 2.12. In Vivo Protection Assay in Hamster Model

Syrian hamsters were obtained from the SPF animal facility of the Institute of Cytology and Genetics SB RAS (Novosibirsk, Russia). Animals were weighed before the experiment and daily during the experiment. Both prophylactic and therapeutic schemes were used to assess the protective efficacy of Nb-Fc homodimers. In the prophylactic scheme, three animal groups (*n* = 5) were intraperitoneally (i/p) administered with 10 mg/kg H5-Fc, 10 mg/kg G7-Fc, 10 mg/kg total human IgG (negative control) 24 h before infection with the virus isolate SARS-CoV-2/human/RUS/Nsk-FRCFTM-1/2020 (EPI_ISL_481284) characterized by a single D614G substitution in the S protein (lineage B.1). The infection was performed intranasally (50 μL/nostril) at a total dose of 1 × 10^4^ plaque-forming units. In the therapeutic regimen, one group of animals (*n* = 5) was infected as above and administered H5-Fc (10 mg/kg, i/p), 10 mg/kg total human IgG 6 h post-infection. The second experiment used a prophylactic group (*n* = 4) 10 mg/kg 2F4-Fc (−1 dpi) and two treatment groups (*n* = 5) receiving 10 mg/kg 2F4-Fc. The viral dose was the same. Hamsters were euthanized on day 5 post-infection. Viral load was quantified in extracted lungs, and antibodies were tested in the blood [21].

### 2.13. Statistical Analyses

Statistical analysis of the significance of differences between groups was performed in GraphPad Prism 8: two-way ANOVA (multiple comparisons) for the weight change analysis, ordinary one-way ANOVA (multiple comparisons) for the viral load analysis, and unpaired t-test for amino acid substitution analysis.

## 3. Results

Llama was immunized with recombinant RBD of the Wuhan variant of SARS-CoV-2. The titer of RBD-specific antibodies after the fourth immunization was 1:50,000. A phage library, denoted as library 1, was constructed using the mRNA from leukocytes of the immunized animal (see Figure 1A) for the immunization scheme and clone count results. RBD-specific nanobodies in this library were identified by a two-step panning strategy, which allowed the selection of phages competing with RBD for binding to immobilized hACE2 [28]. The specificity of the selected phages was assessed using ELISA, followed by Sanger sequencing of the nanobody CDS. Following expression in *E. coli* and affinity purification, two surrogate methods were used to assess the potential neutralizing activity of the nanobodies obtained. The first method involved a flow cytometry test to assess the ability of nanobodies to block the binding of fluorochrome-labeled RBD to hACE2 expressed on the surface of HEK293T cells (Appendix A). The second method was ELISA, in which nanobodies competed with hACE2 immobilized on the surface of 96 well plates for binding with RBD of the Wuhan viral variant (Appendix A). Out of the 20 selected clones with unique sequences, 16 showed neutralizing activity in these assays. However, only three of them, namely G7, H5, and 2F4, were able to completely block the binding of labeled RBD to ACE2-expressing cells at a concentration of 2 µg/ml. These three clones were selected for further analysis.

Utilizing biolayer interferometry (BLI), it was determined that H5 bound to RBD with a KD below 1.0 × 10^−12^ M, which is related to extremely slow off-rate, whereas G7 and 2F4 exhibited KD values of 1.72 × 10^−10^ M and 3.87 × 10^−10^ M, respectively (Figure 1B). BLI was also employed to assess the competition between the three nanobodies for binding to the RBD of the Wuhan Spike protein. RBD was immobilized on the sensor, and one of the antibodies was added. Following the formation of a complex, a second antibody was added to see if additional sensor saturation could be achieved. H5 and G7 completely blocked each other’s interaction with RBD, indicating a significant overlap in the epitopes they recognized (see Figure 1C). In contrast, 2F4, when paired with either of these antibodies, exhibited independent binding to RBD.

To evaluate the in vitro neutralizing properties of the selected nanobodies, lentiviruses pseudotyped with the Spike protein of the Wuhan variant of SARS-CoV-2 (sVP) were used. Monomeric forms of H5, G7, and 2F4 demonstrated the ability to neutralize sVP with IC_50_ values of 0.94 nM (15.92 ng/mL), 1.24 nM (20.64 ng/mL), and 3.52 nM (58.13 ng/mL), respectively. Dimeric forms of H5, G7, and 2F4, fused to the human IgG1 Fc fragment, neutralized sVP with IC_50_ values of 0.14 nM (11.06 ng/mL), 0.13 nM (10.58 ng/mL), and 0.39 nM (30.86 ng/mL), respectively (see Figure 2A).

The emergence of new variants of SARS-CoV-2 necessitated an evaluation of the breadth of neutralization of the obtained nanobodies. To address this question, we constructed a panel of lentiviruses pseudotyped with the S proteins of relevant variants circulating in 2021–2022, such as Delta, Lambda, Kappa, Omicron BA.1 and BA.5. Nanobody G7 neutralized the Delta variant approximately 5 times less effectively than the Wuhan variant and exhibited no activity against other variants. H5, in addition to Wuhan, neutralized Delta and Kappa variants but not Lambda or Omicron variants. 2F4 neutralized all pre-Omicron evolutionary variants of SARS-CoV-2 with similar efficacy but did not show activity against the Omicron lineage variants (Figure 2B).

The protective properties of these nanobodies (tested as VHH-Fc dimers) were examined using a Syrian hamster model. Both prophylactic and therapeutic regimens were employed (Figure 3A). In the prophylactic regimen, antibodies were administered intraperitoneally one day before infection with the Wuhan variant virus at a dose of 10 mg/kg body weight. The control group of animals received 1 mg of human total IgG. In the therapeutic regimen, hamsters received nanobodies at the same dose of 10 mg/kg body weight six hours after infection. Animals were euthanized on the fifth-day post-infection. Blood and lungs were collected from euthanized animals. The protective effect of the antibodies was assessed based on the dynamics of weight change and the quantity of viral RNA copies in lung tissue homogenates.

Prophylactic and therapeutic administration of all three nanobodies showed their significant protective activity during the course of infection. While control animals lost 12–20% of their weight by the fifth day of infection, experimental animals displayed a positive weight dynamic starting from the third day (Figure 3B). QPCR-based quantification of viral genomic RNA in lung homogenates revealed that hamsters receiving prophylactic or therapeutic antibody injections had 3 to 6 orders of magnitude less viral RNA compared to control animals that received a placebo in the form of human total IgG (Figure 3C).

In an attempt to identify nanobodies with a broader spectrum of activity, additional panning within the phage library 1 was performed using RBD from the Omicron BA.1 variant as the target. As a result, 12 unique nanobody-encoding clones were selected. Phylogenetic analysis revealed that eight of them had highly similar sequences, including shared CDR3 with the nanobodies obtained in the first screening, which had weak neutralizing activity (Figure 4A). Four discovered clones with CDR3 sequences not found in the first screening encoded nanobodies lacking any neutralizing activity based on the results of a surrogate flow cytometry-based neutralization assay. These results indicated that nanobodies recognizing conserved RBD epitopes yet possessing high SARS-CoV-2-neutralizing activity were absent or relatively rare in the llama immune repertoire after immunization with the Wuhan RBD variant.

To increase the frequency of broadly neutralizing nanobodies, we conducted additional immunization of this animal with two injections of equimolar mixtures of Beta and Delta RBD variants. Subsequently, with a two-month interval, three injections of RBD from SARS-CoV-1 were given, followed by four injections of the Omicron BA.5 variant after a six-month break (see Figure 1A). This immunization regimen was not rationally planned. Rather, that was an attempt to follow the emergence of new virus variants. In total, this llama received 13 antigen injections over a period of two and a half years. After the final immunization, phage library VHH #2 was constructed. This library was first enriched with clones binding to RBD of the BQ.1 variant; then, in the second stage, RBD of the XBB.1.5 variant was used as the target, and in the third stage, enrichment was carried out with clones displacing hACE2 from the complex with RBD-XBB1.5. As a result of this search, six nanobodies with unique CDR3 sequences were selected out of 96 tested clones. According to the results of phylogenetic analysis, only one of them, 9A7, was closely related in sequence to library 1-derived clones A8, H7, and C6 (Figure 4A). The CDR3 sequences of 9A7 vs. A8 and H7 differ by only two amino acid residues, which is consistent with their common origin. In the VH segment, 9A7, A8, and H7 differ from their putative germline ancestor by 17, 2, and 2 amino acid residues, respectively. Therefore, it can be concluded with a high degree of certainty that 9A7 is the result of the somatic maturation of one of the nanobodies induced by the Wuhan variant of RBD. The remaining clones from library 2 were structurally unrelated to the clones from library 1.

Library 2-derived nanobodies 9A4, 9A7, 9A21, 9A57, 9A58, and 9A80 were expressed and tested for neutralizing activity against a panel of pseudoviruses carrying S proteins of the Wuhan, BA.5, BQ.1, and XBB.1 variants. Nanobody 9A80 showed no neutralizing activity, and 9A4 neutralized the Wuhan, BA.5, and BQ.1 variants but not XBB.1. The nanobodies 9A7, 9A21, 9A57, and 9A58 neutralized all four tested pseudoviruses with IC_50_ values ranging from 0.5 to 29.4 nM (Figure 5A).

Nanobodies 9A7, 9A21, 9A57 and 9A58 were then produced in a dimeric form as fusions with Fc fragment of human IgG1 and their neutralizing activity was tested against a broader panel of pseudoviruses encompassing Wu, BA.1, BA.1.1, BA.2, BA.5, BQ.1, BQ.1.1, XBB, XBB.1, XBB.1.5, and XBB.1.16 (Figure 5A). All four Fc-dimers were capable of neutralizing all the variants of pseudoviruses tested. 9A7-Fc had the highest activity against the Wuhan variant at IC_50_ = 1 nM (88 ng/mL), whereas the rest of the variants were neutralized at IC_50_ ranging from 1.4 to 13.1 nM (117 to 1107 ng/mL). 9A21-Fc displayed the strongest activity against BA.1.1 (IC_50_ = 0.8 nM or 66 ng/mL), with the rest of the variants neutralized at IC_50_ ranging from 1.2 to 18.9 nM (100 to 1539 ng/mL). 9A57-Fc was relatively weakly neutralizing against the ancestral Wuhan variant (IC_50_ = 7.9 nM or 636 ng/mL), yet it demonstrated an ultrapotent neutralizing activity against BA.5, BQ.1, and BQ.1.1 (IC_50_ = 0.01, 0.04 and 0.05 nM or 1, 3, and 4 ng/mL) and high activity against XBB lineage variants (IC_50_ = 0.4–1.7 nM or 32–100 ng/mL). 9A58-Fc could neutralize the pseudoviruses tested with IC_50_ values from 0.46 to 17.7 nM (38 to 1468 ng/mL).

Next, BLI was used to compare the dissociation constants of complexes formed by these four broadly neutralizing nanobodies and the non-neutralizing nanobody 9A80 with RBDs from Wuhan, BA.1, BA.5, and XBB.1.5 variants. All nanobodies, including 9A80, bound the RBD variants with affinities in a nano- to picomolar range (KD = 1.15 × 10^−9^ < 1.0 × 10^−12^ M). 9A7 has a higher affinity to RBDs from Wuhan, BA.1, and BA.5 variants (KD = 1.37 × 10^−10^–4.07 × 10^−10^ M) and a lower affinity to RBD from XBB.1.5 (KD = 2.39 × 10^−9^ M). Nanobodies 9A21, 9A57 and 9A80 showed higher affinity to Wuhan and BA.1 RBDs (KD = 2.8 × 10^−10^–7.79 × 10^−10^ M) but were of less affinity to RBDs from BA.5 and XBB.1.5 variants (KD = 1.15 × 10^−9^–3.19 × 10^−9^ M). 9A58 bound RBD from Wuhan and XBB.1.5 with an affinity at least 60–95 times lower than BA.1 and BA.5, yet even for these variants, the affinities remained in the picomolar range (Figure 5B).

Competition of the dimeric Fc-fusions of the broadly neutralizing nanobodies 9A7-Fc, 9A21-Fc, 9A57-Fc, and 9A58-Fc for binding to RBD Wu was assessed by BLI (see Figure 6A). The sensograms obtained demonstrate independent binding of 9A7 and 9A57 from each other and from 9A21 and 9A58. The pre-incubation of 9A58 with RBD blocked the binding of 9A21; however, in the reverse set-up, 9A21-RBD complex formation only partially impeded the binding of 9A58. Furthermore, the stepwise addition of 9A57, 9A7, 9A21, and 9A58 monomers to RBD allowed the binding of all four nanobodies (Figure 6B). These results indicate that these nanobodies can be used as building blocks to construct bi- and oligo-specific antibodies.

## 4. Discussion

The emergence of SARS-CoV-2 mutant variants has posed a significant challenge to the developed anti-coronaviral armamentarium. Despite the pandemic officially ending, tens of thousands of new infections are reported worldwide daily, and the possibility of seasonal outbreaks remains. The spread of resistant virus variants, especially in the absence of therapeutic antibodies, poses a particular threat to individuals at risk, especially those with various forms of immunodeficiency. In such cases, infection can lead to severe disease and even death. Consequently, the identification of antibodies capable of neutralizing new SARS-CoV-2 variants remains a pressing concern. Of special interest are the broadly neutralizing antibodies that could resist future virus variants and prevent the formation of resistant viral variants during antiviral therapy in immunodeficient patients.

The objective of this study was to evaluate the breadth of neutralization of nanobodies in the immune repertoires of llamas at different immunization steps. Two phage libraries encoding llama nanobodies were obtained and panned: the first library was generated based on the repertoire induced by immunization with the RBD of the Wuhan variant of SARS-CoV-2, and the second library was based on the repertoire of the same animal following prolonged immunization with RBDs from SARS-CoV-2 viral variants, including Beta, Delta, and Omicron BA.5, as well as the RBD of the related SARS-CoV-1 Spike protein. Screening library 1 yielded three highly potent nanobodies. When reformatted into a bivalent chimeric form having a human IgG1 Fc fragment, these nanobodies effectively protected Syrian hamsters from the Wuhan variant of SARS-CoV-2. However, none of these antibodies displayed an appreciable breadth of neutralization or the ability to block Omicron lineage variants. Using RBD from the Omicron BA.1 variant as a target for re-screening the same library 1, we demonstrated that the repertoire contained nanobodies capable of binding the conserved RBD epitopes. Nevertheless, none of the identified BA.1-binding nanobodies exhibited potent neutralizing activity. The majority of the identified BA.1-specific clones were found to be closely related in sequence to the previously discovered low-neutralizing nanobodies.

The results of screening the library 2 was significantly different. Among the 96 phages taken from the enriched library, we identified six clones with unique VHH insert sequences in the CDR3 region. Five of them displayed neutralizing activity, and four clones, namely, 9A7, 9A21, 9A57, and 9A58, potently neutralized various tested virus variants, including BQ.1 and four XBB lineage variants. Nanobody 9A80 did not exhibit neutralizing activity but bound RBD variants of Wuhan, BA.1, BA.5, and XBB.1.5 with affinities in the nanomolar and picomolar ranges. Importantly, competitive binding analysis using BLI revealed that the four broadly neutralizing nanobodies could bind the RBD simultaneously, i.e., they recognized non-overlapping or partially overlapping epitopes.

Our findings demonstrate that the immune repertoire induced by the Wuhan variant RBD consists primarily of nanobodies with limited to low neutralization breadth. While it is possible that more comprehensive screening campaigns might identify antibodies with higher activity against omicron variants, like in other studies [16,18,19], the fraction of such nanobodies is apparently low. We believe that the primary reason for the substantial difference in screening results for the two libraries is the predominant stimulation of antibodies against conserved epitopes during multiple contacts of the llama’s immune system with different RBD variants. According to the results of the phylogenetic analysis of nanobody sequences and the estimates of the percentage of amino acid substitutions, one of the library 2 nanobodies, 9A7, is inferred to have evolved through prolonged somatic maturation following RBD immunization. We cannot definitively pinpoint at which stage of immunization the other identified clones from library 2 were induced, but the high level of differences from the known genomic VHH sequences of the llama genes (15–31% substitutions) also suggests their lengthy maturation (Figure 4B).

A significant outcome of our study is the finding that there are multiple conserved vulnerability sites in the RBD structure. Only a handful of the thousands of neutralizing human SARS-CoV-2-specific antibodies described over the past three years are known to possess activity against a broad spectrum of viral variants [29]. The discovery of four nanobodies exhibiting such activity in one small-scale screening campaign is remarkable. Based on the altered affinity and potency of the four nanobodies against various viral variants, one can conclude that the mutation drift of SARS-CoV-2 clearly affects the structure of the epitopes recognized. Notwithstanding, the magnitude of the effect appears limited, and no complete loss of neutralizing activity was observed. It is possible that this result is due to the peculiarities of nanobody epitopes and the preservation of their structure in the context of viral evasion under the pressure of conventional antibodies. To draw more definite conclusions, further research is required, including an analysis of the three-dimensional structure of nanobody/Spike complexes compared to that of the broadly neutralizing human antibodies and the study of the competition between antibodies of different formats in binding to RBD or the coronavirus Spike protein. From a practical perspective, the presence of multiple conserved “nanoepitopes” offers opportunities for constructing next-generation bi- and oligo-specific antiviral agents potentially capable of neutralizing current and future virus variants while blocking the formation of resistant viruses during antibody prophylaxis/therapy in immunodeficient patients.

## Figures and Tables

**Figure 1 vaccines-12-00129-f001:**
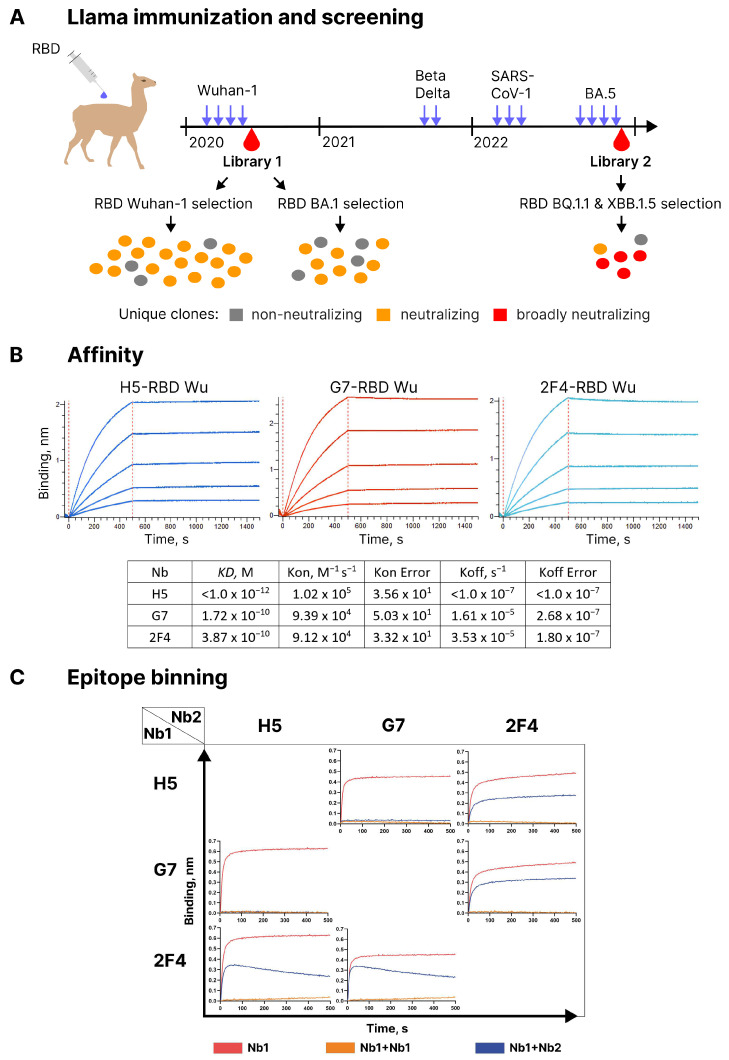
(**A**) Outline of serial immunization of a llama by various SARS-CoV-2 RBD variants, as well as the screening results. (**B**) BLI analysis of the interaction between nanobodies H5, G7, and 2F4 with the RBD of SARS-CoV-2 Wuhan. (**C**) Competition BLI assay for nanobodies H5, G7, and 2F4 binding to RBD-His6 Wu.

**Figure 2 vaccines-12-00129-f002:**
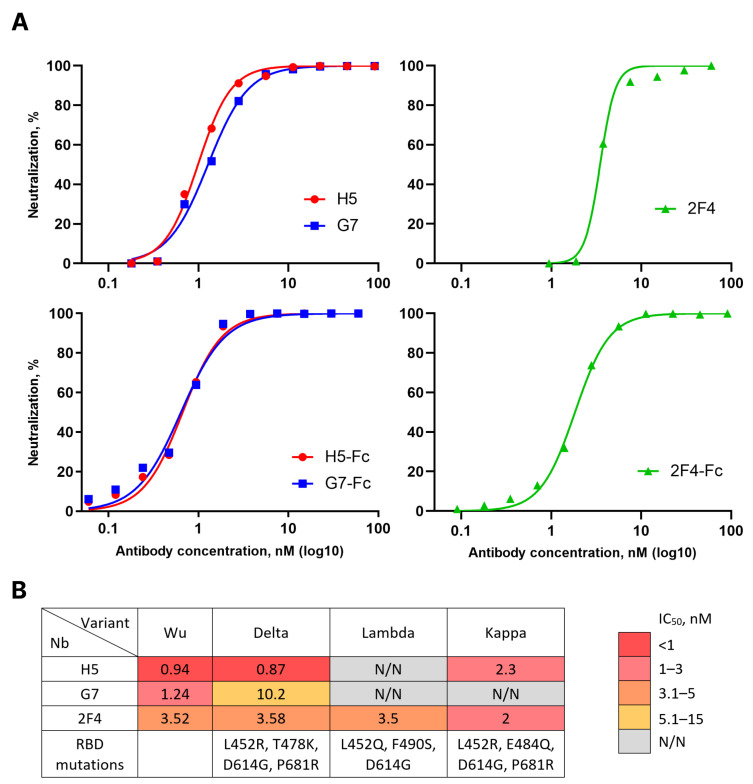
(**A**) Activity of H5, G7, and 2F4 nanobodies (Nbs) and nanobody-Fc fusions in pseudovirus neutralization assay with Wuhan variant of the SARS-CoV-2 Spike protein. Data from four independent experiments are presented. (**B**) Activity of H5, G7, and 2F4 Nbs in pseudovirus neutralization assay using Wuhan, Delta, Lambda, and Kappa variants of the SARS-CoV-2 Spike protein. N/N—no visible neutralization.

**Figure 3 vaccines-12-00129-f003:**
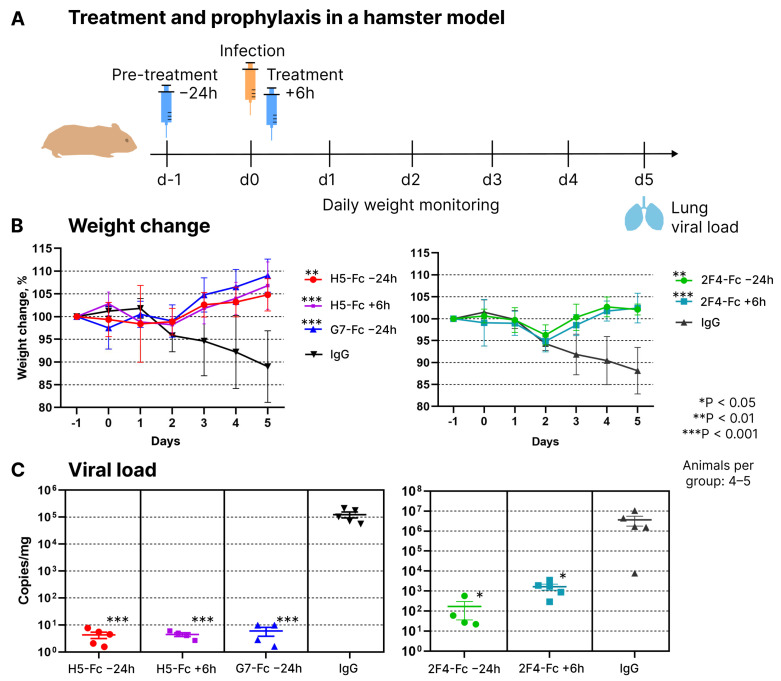
(**A**) Scheme of using H5, G7, and 2F4 Fc-fusions in treatment and prophylaxis of SARS-CoV-2 infection in a hamster model. (**B**) Relative weight dynamics in the prophylaxis and treatment groups following administration of Nb-Fc or control total human IgG. Error bars represent means ± SD. (**C**) Impact of Nb-Fc on the levels of viral transcripts in hamster lungs (RT-qPCR with *RdRp*-specific primers). RT-qPCR with *E* gene-specific primers produced very similar results (not shown). Error bars represent means ± SEM.

**Figure 4 vaccines-12-00129-f004:**
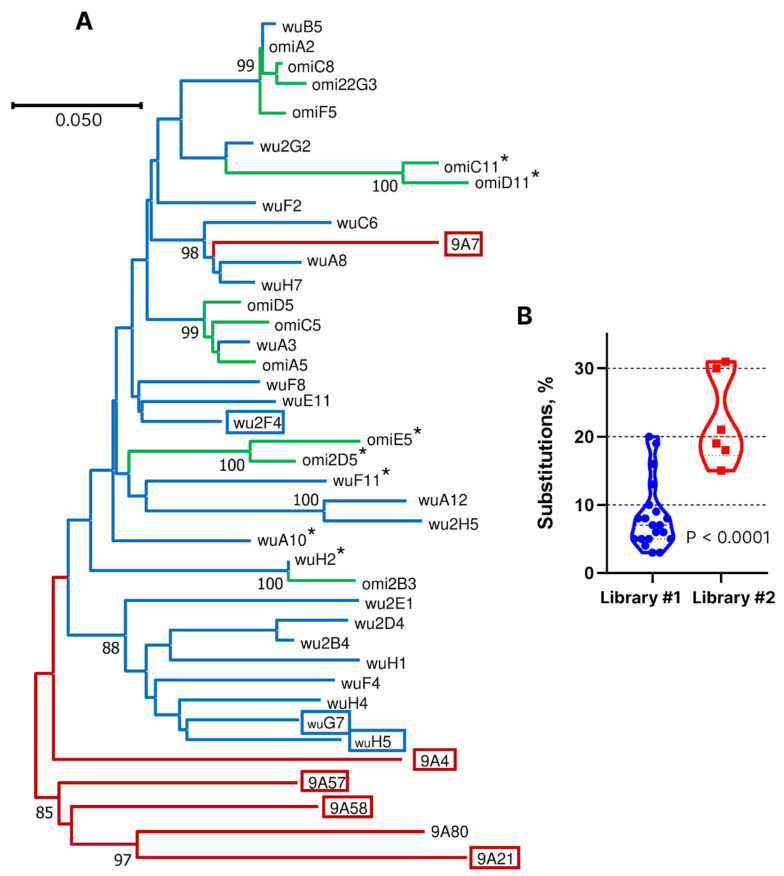
(**A**) Dendrogram of similarity of nanobody amino acid sequences. Potent neutralizing nanobodies are boxed, and those devoid of neutralization activity are marked with an asterisk. Blue and green correspond to library 1 (Wuhan and Omicron), and red are from library 2. (**B**) Percentage of amino acid residue substitutions in the identified nanobodies compared to the germline-derived sequences.

**Figure 5 vaccines-12-00129-f005:**
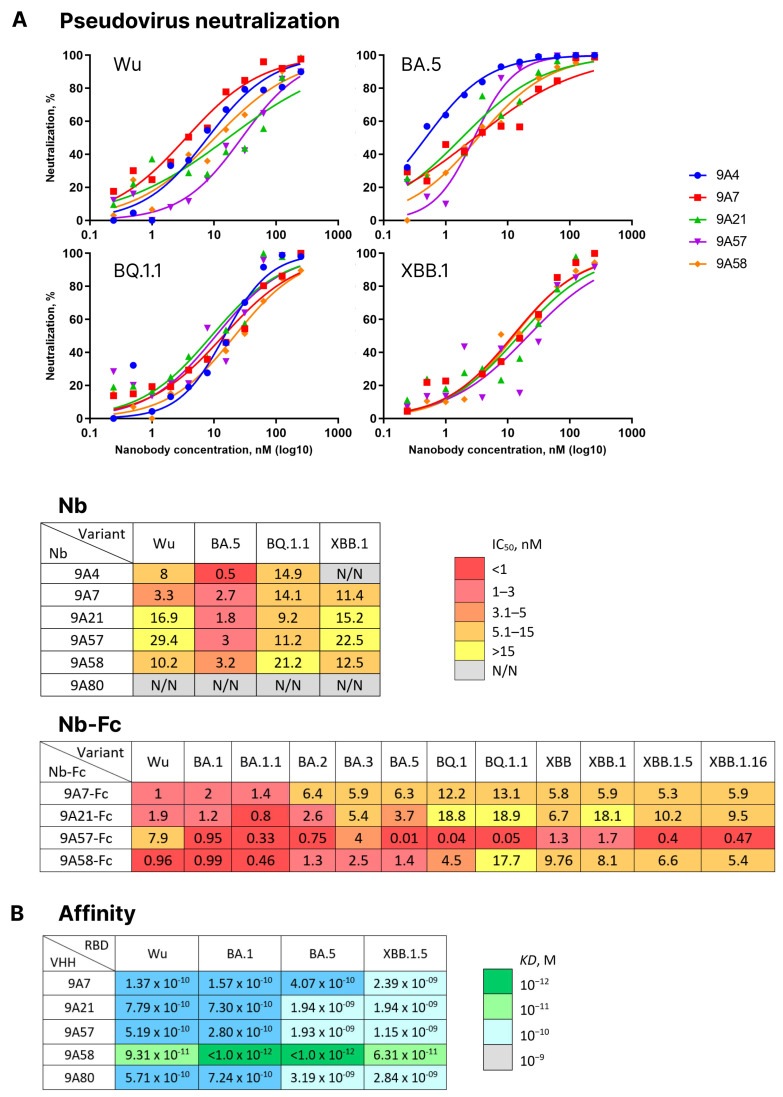
(**A**) Activity of nanobodies (Nb) 9A4, 9A7, 9A21, 9A57, and 9A58 in pseudovirus neutralization assay using Wuhan, BA.1, BA.5, and XBB.1 SARS-CoV-2 Spike variants, as well as the activity of 9A7, 9A21, 9A57, 9A58 Fc-fusions (Nb-Fc) in pseudovirus neutralization assay using Wuhan, BA.1, BA.1.1, BA.2, BA.5, BQ.1, BQ.1.1, XBB, XBB.1, XBB.1.5, and XBB.1.16 variants. (**B**) Affinity values for the interaction of 9A7, 9A21, 9A57, 9A58, 9A80, and RBDs of Wuhan, BA.1, BA.5, and XBB.1.5 SARS-CoV-2 variants.

**Figure 6 vaccines-12-00129-f006:**
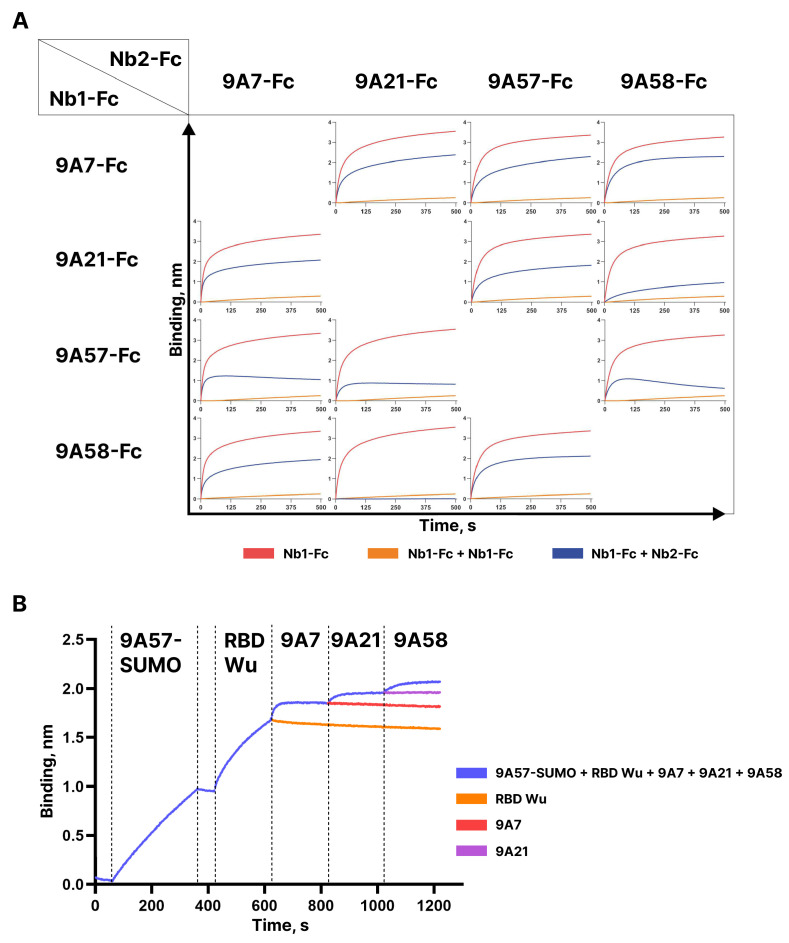
(**A**) BLI analysis of competition between 9A7-, 9A21-, 9A57-, 9A58-Fc fusions for binding the RBD-His6. (**B**) BLI analysis of competition between Nbs, including 9A57-SUMO fusion binding the RBD Wu.

## Data Availability

Data supporting reported results can be requested from the corresponding author.

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
