# Peer review of "Serial Llama Immunization with Various SARS-CoV-2 RBD Variants Induces Broad Spectrum Virus-Neutralizing Nanobodies"

_vaccines, 2024, doi:10.3390/vaccines12020129_

Round 1
Reviewer 1 Report
Comments and Suggestions for Authors
This study aims to compare 19 nanobody repertoires raised in the same llama after immunization with Wuhan variant RBD and 20 after subsequent serial immunization with a variety of RBD variants including that of SARS-CoV-1.
Overall, I find this manuscript is very interesting. The experiment has been well organized by the authors. The novel nanobodies identified in this study may contribute to the neutralizing antibodies development for human Coronavirues diseases. The affinity is acceptable and still can be improved since the clones have been obtained. Great findings.
Author Response
We thank the reviewer for their time and expertise.
Reviewer 2 Report
Comments and Suggestions for Authors
The manuscript titled "Serial Llama Immunization with Various SARS-CoV-2 RBD Variants Induces Broad Spectrum Virus-Neutralizing Nanobodies" evaluates the breadth of neutralization of nanobodies produced from a llama against SARS-CoV-2. The manuscript demonstrates the potential therapeutic application of the nanobodies using a hamster challenge model. The manuscript also provides an evaluation of the nanobodies against the ever growing problem of VOCs. Overall, the manuscript is well written and organized. The introduction provides an excellent review on nanobodies and their potential application. The methods include great detail. The discussion is supported by the results. Lastly, the figures and tables are clear. This is an interesting study on nanobodies and its clear this therapy does have applications in combating SARS-CoV-2 infections.
I have only minor comments to share:
Line 208: Add space between pH8.0
Line 365: What is meant by "One group of hamsters remained intact"? Please clarify.
Author Response
We thank the reviewer for their time and expertise.
Comments 1: Line 208: Add space between pH8.0.
Response 1: – Corrected.
Comments 2: Line 365: What is meant by "One group of hamsters remained intact"? Please clarify.
Response 2: – There was no intact group in this experiment. We thank the Reviewer for pointing out this mistake. The sentence was removed.
Reviewer 3 Report
Comments and Suggestions for Authors
In the article titled "Serial Llama Immunization with Various SARS-CoV-2 RBD Variants Induces Broad Spectrum Virus-Neutralizing Nanobodies," the authors assessed the neutralization capacity of nanobodies generated in a llama at different stages of immunization against the RBD of SARS-CoV-2 variants. Two phage libraries were generated with llama nanobodies. The first library was obtained after immunizing a llama with the RBD of the Wuhan variant of SARS-CoV-2, and the second after extended immunization with RBDs of SARS-CoV-2 variants, including Beta, Delta, and Omicron BA.5, as well as the RBD of the Spike protein of SARS-CoV-1. Following the screening of the first phage library, three potent nanobodies were identified, which, when reformatted as bivalent chimeras, effectively protected hamsters against the Wuhan variant. However, these nanobodies showed limited capacity to neutralize Omicron lineage variants.
From the second library, four clones (9A7, 9A21, 9A57, and 9A58) were identified as capable of neutralizing various SARS-CoV-2 variants, including BQ.1 and four XBB lineage variants. The four broadly neutralizing nanobodies could simultaneously bind to the RBD, recognizing non-overlapping epitopes. The results demonstrate the feasibility of generating nanobodies capable of neutralizing diverse SARS-CoV-2 variants through serial immunizations with RBDs from different SARS-CoV-2 variants. These nanobodies hold potential as prophylactic and therapeutic tools for immunocompromised patients against various SARS-CoV-2 variants.
While the article is well-written with clear objectives, and the materials and methods employed are generally appropriate and thorough in addressing the stated objectives, it requires modifications to enhance its quality.
Summary of suggested modifications:
Inclusion of Immunization Protocol Approval:
Include in the Materials and Methods section the approval of the immunization protocol in hamsters and llamas by the relevant institution.
Review of Figure 1A:
Correct the representation of neutralizing clones drawn after selection with RBD BA1 in Figure 1A, as per the text indicating no neutralizing clones were found for RBD BA1 in library number 1.
Review of Paragraph in Lines 384-392:
Verify and correct the paragraph mentioning pseudovirus neutralization results for Beta, Omicron, BA.1, and BA.5 variants of the Spike protein. Additionally, clarify the absence of Figure 2C.
Clarification in Figure 2 Caption:
Add to the caption of Figure 2 that the S protein refers to the Spike protein of the viral variants expressed in the pseudoviruses used in the study.
Combining Results in Figure 2A:
Combine in one graph for each viral variant the results of neutralization by monovalent and bivalent nanobodies in Figure 2A for better result comparison.
Additional Data in Figure 3B:
Add weight change and viral load data for animals treated with G7 (G7 6H) in Figure 3B. Also, clarify if the results shown for pre-treatment and treatment with 2F4 correspond to the monomeric or bivalent form (2F4-Fc) of the nanobody.
Clarification in Lines 446-447:
Correct the sentence "In the case of 9A7, 9A21, 9A57, and 9A80, the interaction affinities with different RBD variants varied within an order of magnitude" to accurately reflect differences in interaction affinities for each nanobody and viral variant tested. Provide more detailed commentary in the text on these differences.
These modifications will improve the clarity, precision, and overall presentation of the article before publication.
Author Response
We thank the reviewer for their time and expertise.
Comments 1: Inclusion of Immunization Protocol Approval
Include in the Materials and Methods section the approval of the immunization protocol in hamsters and llamas by the relevant institution.”
Response 1: According to the journal rules qnd author’s guidelines, the approval is described in the section “Institutional Review Board Statement”
Comments 2: Review of Figure 1A:
Correct the representation of neutralizing clones drawn after selection with RBD BA1 in Figure 1A, as per the text indicating no neutralizing clones were found for RBD BA1 in library number 1.
Response 2: There was an inaccuracy in the text. In fact, there were 8 weakly neutralizing, 4 non-neutralizing. Corrected (Line 402 in the revised version).
Comments 3: Review of Paragraph in Lines 384-392:
Verify and correct the paragraph mentioning pseudovirus neutralization results for Beta, Omicron, BA.1, and BA.5 variants of the Spike protein. Additionally, clarify the absence of Figure 2C.
Response 3: The data in the paragraph and Fig. 2B were corrected. The designation 2C was erroneous and was corrected to 2B – (Lines 356-364 in the revised version).
Comments 4: Clarification in Figure 2 Caption:
Add to the caption of Figure 2 that the S protein refers to the Spike protein of the viral variants expressed in the pseudoviruses used in the study.
Response 4: For the sake of uniformity, “S-protein” was replaced by “Spike protein” throughout the text.
Comments 5: Combining Results in Figure 2A:
Combine in one graph for each viral variant the results of neutralization by monovalent and bivalent nanobodies in Figure 2A for better result comparison.
Response 5: The graphs cannot be combined as they represent independent experiments with different antibody titrations.
Comments 6: Additional Data in Figure 3B:
Add weight change and viral load data for animals treated with G7 (G7 6H) in Figure 3B. Also, clarify if the results shown for pre-treatment and treatment with 2F4 correspond to the monomeric or bivalent form (2F4-Fc) of the nanobody.
Response 6: The G7-Fc antibody was tested only in the prophylactics regimen. Results are for 2F4-Fc, corrected.
Comments 7: Clarification in Lines 446-447:
Correct the sentence "In the case of 9A7, 9A21, 9A57, and 9A80, the interaction affinities with different RBD variants varied within an order of magnitude" to accurately reflect differences in interaction affinities for each nanobody and viral variant tested. Provide more detailed commentary in the text on these differences.”
Response 7: Described in more detail (Lines 460-464 in the revised version).
Reviewer 4 Report
Comments and Suggestions for Authors
The authors describe efficient isolation of SpikeRBD targeting nanobodies (Nbs) from llama, using immunization and phage display and screening. Nbs isolated after various immunization campaigns are compared and described with their antigen affinity, and neutralization potency towards different variants of RBD. For certain clones (resulting from Wuhan-RBD immunization campaign only), protective effect in vivo is also shown. Importantly, broadly neutralizing variants could be isolated after immunization campaigns using first Wuhan-RBD and later other RBD variants. Interestingly, one clone discovered by screening the Nb library created with this strategy could bind to all tested variants, but had no neutralizing activity in the pseudotyped virus neutralization assay. Other Nbs discovered were broadly neutralizing, revealing different potency for different virus variants. Epitope binning (competition) was also performed and most of the isolated Nbs were non-competing, so they could constitute parts of multispecific reagents contributing to future antiviral therapeutic agents. Although there are very many similar reports, the findings presented here could in this view support pandemic preparedness.
The manuscript is clearly written and the rationale of the study easy to follow (except for the immunization strategy, which the authors say themselves). Certain passages should be reorganized (please see the remarks below). In data presentation, the deviations should be explicit as the authors have performed parallel experiments. Reference list (order of references) is not OK (e.g. 26 should be describing RNA isolation protocol and appears in the bibliography as Amanat et. al (serological assays for SARS/CoV2), this is why
Please find below a list of remarks which I hope will support the revision.
Line 70: please compare: doi: 10.1016/j.isci.2023.107085.
Line 168: ug should be microgram symbol (please correct throughout the text, and also the ul – units)
Line 168: centrifugation conditions should be in g units, not rpm
Line 170: NaHCO3, 3 in subscript (please correct throughout the text), and please specify the pH
Line 192-195: please specify the antibody identity with RRIDs (or catalogue numbers if no RRID available)
Line 196: is there a reaction quenching step present (like acid addition, or similar)?
Line 230: source of SUMO protease and concentration in the reaction?
Line 324: Supplementary Figure 1A is cited in text in front of S1B
Supplementary Figure 1B. Please consider using the nanomolar units for the concentration of nanobody
Line 333: “below the detection threshold” – I suppose the authors try to describe the extremely slow off-rate, please reword. In addition, surely the isolated Nbs are of high affinity, but to make precise comments on the values another method (eg. Competition analysis) should be used, because at the moment the dissociation rate iss too slow judging from BLI plots.
Figure 2A: Error bars are missing. Please consider showing all Nbs in one plot and all Nb-Fcs in a second plot. It would also be helpful to stick to the same colors and symbols for the same Nb protein.
Figure 2B: please use nanomolar units.
Figure 2B legend: Figure is not cited in the text. Deviations are not given (In Materials and Methods section, 2 parallel set-ups are described). Please indicate are these experiments done with Nbs or Nb-Fcs.
Lines 384-392: This paragraph refers to Figure 2B (there is no Figure 2C) and should come in front of the Description of Figure 3 (either the Figure 2B is not complete or 2C is missing)
Lines 404-423: Phylogenetic relatedness and germline deviations are very interesting information, but the listing of sequences (even as Supplementary Material) would make this passage easier to read, and importantly, the clones discovered here could be compared to those found in similar studies.
Figure 5A. Please use molar units for concentration, because of the different molecular size of the proteins used. Legend to 5A plots: complete names of the clones should be given. Error bars are missing.
Author Response
We thank the reviewer for their time and expertise
Reference list was corrected. Thank you for pointing this out.
Comments 1: Line 70: please compare: doi: 10.1016/j.isci.2023.107085.
Response 1: While the mentioned article provides interesting results, it does not contain information on the ability of the nanobody to neutralize XBB variants of SARS-CoV-2.
Comments 2: Line 168: ug should be microgram symbol (please correct throughout the text, and also the ul – units).
Response 2: Corrected throughout the text.
Comments 3: Line 168: centrifugation conditions should be in g units, not rpm.
Response 3: Corrected.
Comments 4: Line 170: NaHCO3, 3 in subscript (please correct throughout the text), and please specify the pH.
Response 4: Corrected, pH specified.
Comments 5: Line 192-195: please specify the antibody identity with RRIDs (or catalogue numbers if no RRID available).
Response 5: RRIDs or catalogue numbers were added.
Comments 6: Line 196: is there a reaction quenching step present (like acid addition, or similar)?
Response 6: Yes, added.
Comments 7: Line 230: source of SUMO protease and concentration in the reaction?
Response 7: Added.
Comments 8: Line 324: Supplementary Figure 1A is cited in text in front of S1B.
Response 8: Corrected.
Comments 9: Supplementary Figure 1B. Please consider using the nanomolar units for the concentration of nanobody.
Response 9: Сonverted.
Comments 10: Line 333: “below the detection threshold” – I suppose the authors try to describe the extremely slow off-rate, please reword. In addition, surely the isolated Nbs are of high affinity, but to make precise comments on the values another method (eg. Competition analysis) should be used, because at the moment the dissociation rate is too slow judging from BLI plots.
Response 10: We agree with the Reviewer that the expression was indeed incorrect. Reworded.
Comments 11: Figure 2A: Error bars are missing. Please consider showing all Nbs in one plot and all Nb-Fcs in a second plot. It would also be helpful to stick to the same colors and symbols for the same Nb protein.
Response 11: The results presented are from independent experiments with different antibody titrations, which prevents us from combining them. Color scheme and symbols were corrected.
Comments 12: Figure 2B: please use nanomolar units.
Response 12: Corrected.
Comments 13: Figure 2B legend: Figure is not cited in the text. Deviations are not given (In Materials and Methods section, 2 parallel set-ups are described). Please indicate are these experiments done with Nbs o rNb-Fcs.
Response 13: We mentioned that the activity of Nbs and Nb-Fc was tested in independent experiments. The Mat&Met section was corrected.
Comments 14: Lines 384-392: This paragraph refers to Figure 2B (there is no Figure 2C) and should come in front of the Description of Figure 3 (either the Figure 2B is not complete or 2C is missing).
Response 14: Corrected.
Comments 15: Lines 404-423: Phylogenetic relatedness and germline deviations are very interesting information, but the listing of sequences (even as Supplementary Material) would make this passage easier to read, and importantly, the clones discovered here could be compared to those found in similar studies.
Response 15: We agree with the Reviewer that comparison of datasets from different labs would be very interesting but at the moment we cannot show all the sequences because of patent limitations.
Comments 16: Figure 5A. Please use molar units for concentration, because of the different molecular size of the proteins used. Legend to 5A plots: complete names of the clones should be given. Error bars are missing.
Response 16: Units and names were corrected, error bars are not applicable, the experiment was carried out without repeats.
Round 2
Reviewer 4 Report
Comments and Suggestions for Authors
The authors have diligently corrected the manuscript and provided complete and more correct information on materials and methods used, as well as adjusted the logical order in results presentation. I have to ask to review the reference section again (some of the references appear twice, but I am sure purely due to an oversight). With this, I can recommend the manuscript for publication.